# Baseline infection prevalence, risk factors and treatment outcomes of visceral leishmaniasis in Northeastern Uganda: A cross-sectional study

**Patrick Sagaki**[1,2]*, **Jeremiah Mutwalante Twa-Twa**[1], **Posiano Mulalu**[1], **Benon Wanume**[1], **Peter Olupot-Olupot**[1,3]

1 Department of Public Health, Faculty of Health Sciences, Mbale Campus, Busitema University, Mbale City, Uganda, 2 District Health office, Amudat District Local Government, Karamoja region, Moroto City, Uganda, 3 Mbale Clinical Research Institute, Mbale Regional Referral & Teaching Hospital, Mbale City, Uganda

* drsagaki@yahoo.com

**Data Availability Statement:** All relevant data are within the manuscript and its Supporting Information files.

## Abstract

### Introduction

*Visceral leishmaniasis* (VL) also known as Kala-azar is one of the neglected tropical diseases (NTD) of public health importance. Despite being a disease of a long history, the condition remains poorly studied especially in East Africa. For instance, whereas, the geographical location of the disease is known, there is a stark paucity of data on the burden, risk factors and clinical outcomes of this condition in Northeastern Uganda. Therefore, the disease picture in these settings is incomplete. The overarching aim for this study was to describe pre-elimination prevalence, associated factors and treatment outcomes of VL in Moroto District.

### Methods

We conducted a cross-sectional study in which community cases were identified at baseline. They were followed up at Amudat Hospital Kala-azar Treatment Centre for the treatment outcomes. We used a customized data collection tool to elicit data on demographic characteristics, socio-economic and anthropometry. Data were entered on excel database and exported to Stata software for analyses. Proportions and measures of central tendency were computed. Binary associations were determined using Chi-square statistical test. In addition, variables independently associated with VL were determined via logistic regression analyses. At follow up stage, the outcomes were determined.

### Results

The overall prevalence of VL infection in Moroto district was 5.21% (95%CI: 3.15% - 8.48%) with varying county level prevalence at Matheniko, Tepeth and Mororo at 6.90%, 4.49% and 3.61%; respectively. The common risk factors for VL infection included lack of knowledge of habitat for Sand flies, AOR 5.33 (95%CI: 1.69–16.82). Patients with VL presented with

**Funding:** This work was supported by Drugs for Neglected Diseases Initiative (DNDi) to PS to facilitate data collection. The funders had no role in study design, data collection and analysis, decision to publish, or preparation of the manuscript.

**Competing interests:** The authors have declared that no competing interests exist.

fever, headache, abdominal pain and swelling, coughing, night sweats, diarrhea, fatigue, breathlessness, and nose bleeding. The average hospitalization for VL was 17 days. All the patients who were treated at the hospital cured.

## Conclusion

The prevalence of VL in Moroto district was 5.21% and within elimination threshold. The high-risk factors for VL infection included lack of knowledge about the habitat for Sand flies. The average hospitalization for VL was 17 days.

## Author summary

This study investigates the prevalence and risk factors associated with visceral leishmaniasis (VL) in Moroto District, Northeastern Uganda, an area previously underexplored regarding this neglected tropical disease. A cross-sectional approach identified community cases, followed by treatment outcomes monitored at Amudat Hospital. Findings revealed an overall VL prevalence of 5.21%, with county-level variations. Key risk factors included inadequate knowledge about sand fly habitats, significantly impacting infection rates. Patients typically presented with symptoms such as fever, abdominal pain, and fatigue, requiring an average hospitalization of 17 days. The study concludes that while the VL prevalence is within the elimination threshold, public health initiatives must address knowledge gaps about sand fly habitats to reduce infection risk effectively. This research contributes valuable insights into VL epidemiology in East Africa, highlighting the need for targeted interventions in vulnerable communities.

## Introduction

*Leishmaniasis* is a neglected tropical disease (NTD) caused by *Leishmania* parasite. It is transmitted by a bite of infected female phlebotomine sand flies and clinically categorized in three forms, namely:- *Cutaneous leishmaniasis* (CL), *Muco-cutaneous leishmaniasis* (ML) and *Visceral leishmaniasis* (VL) also known as Kala-azar. The disease is treatable and curable with the current efficacious medications, but lethal if left untreated especially Visceral leishmaniasis (WHO). Visceral leishmaniasis poses a serious global health challenge, with approximately 50,000 to 100,000 new cases reported each year, primarily in endemic regions across South Asia, East Africa, and parts of South America, in countries like Brazil, **Ethiopia, Kenya, Somalia, South Sudan, Sudan** and India [1]. Evidence shows that the disease peaks during the sunny and warm pleasant seasons [2]. Some data exist but they have their limitations. For instance, most data are health facility based retrospective studies with their limitations in data quality. Yet other data are limited by the fact that VL has a temporal, geographical and ethnic tendency and so generalization is problematic. Reports have shown that the prevalence of VL in East African region is associated with civil wars, adverse climate patterns, poverty and malnutrition [3], but there remains a dearth in understanding on characteristics of the disease in the pre-elimination stages. For instance, whether the affected populations, disease distribution and its seasonality have changed at the stage of pre-elimination is yet to be formally described. Consequently, targeted interventions lack evidence and may affect elimination of of VL in these settings.

In Uganda *Leishmaniasis* is endemic in North-Eastern region also called Karamoja sub region. In this region, VL is the commonest type in which children under 15 years in a male to female sex ratio of 3:1 are mainly affected. Whereas, some data have indicated its seasonality, in Uganda it is not well documented [4]. Data on the disease in Uganda remains scanty and therefore poorly studied. Having only one treatment center in the country means that patients have to travel long distances to access medical care. Within the context of VL in Northeastern Uganda, we aimed at describing the prevalence, risk factors and outcomes in Moroto District in the pre-elimination era.

## Methods and materials

### Ethics statement

The study proposal was submitted to the Research and Ethics Committee (REC) in Mbale Regional Referral Hospital for approval before data collection. REC approved the study under REC approval number: MRRHREC-OUT-011/2020. After meeting the selected participants to take part in the study, the research team introduced themselves, explained the purpose of the study and sought for written consent from all participants. For child participants, written formal consent was obtained from their parents/guardians. After receiving their consent, the research team administered the questionnaires to the participants.

### Study area

This study was carried out in Moroto district, which is in North-Eastern Uganda. Moroto district is bordered by Kotido District to the north, republic of Kenya to the east, Amudat District to the south, Nakapiripiti District to the southwest and Napak District to the west. Moroto Municipal Council, where the District Local Government Offices are located, is at the foot of Mt. Moroto. Moroto municipal council is approximately 213 kilometers (132 miles) by road, northeast of Mbale town, the nearest large city. This is about 434 kilometers (270 miles), by road, northeast of Kampala, the capital and largest city of Uganda. Moroto district has three counties namely: -Matheniko, Tepeth and Moroto municipality. It has 8 sub counties, 39 Parishes and 217 Villages with a projected total population of 119,249 as of 2020. The major livelihood zones in Moroto district include pastoralism, agro-pastoralism and agriculture with a few people owning businesses in urban areas. The homesteads locally known as "Manyattas" are organized into groups of family units, living within mud walled and grass thatched roofed houses encircled by thorny fencing. Amudat Hospital is the only VL treatment Centre in Uganda. A recent hospital review of VL records in 2014 showed that out of the 255 VL patients treated, 181 (71%) came from Amudat District, 69 from Moroto District, 2 from Kotido district, 2 from Napak district, and 1 from Katakwi district [5]. In Amudat District, the prevalence of VL was estimated as 17.2% in 2013 [6]. Following these findings, concerted efforts have been directed to the control of the disease given its contribution to hospital admissions and the overall health impact on the affected individuals and their communities [7]. There is a tendency of the condition affecting specific communities especially the poor, with low literacy rates [8].

### Study population

The study population were inhabitants of Moroto district at the time of conducting the research.

### Target population

The study targeted individuals aged three months and above within the randomly selected households, within the randomly selected villages and randomly selected Parishes in Moroto district.

### Accessible population

Accessible population consisted of systematically selected individuals from the randomly selected households, within the randomly selected villages and randomly selected Parishes in Moroto district.

### Sample population

This consisted of participants systematically selected within the clusters that fully met the inclusion criteria.

### Study design

This was a cross-sectional study in which community cases were identified and followed up at the hospital for treatment outcome. The baseline information related to prevalence of VL infection, factors associated with VL infection were captured. The participants who turned out positive for VL infection were referred to Amudat Hospital for parasitological diagnosis of VL disease and treatment. The outcome followed included failure to report to Amudat hospital Kala azar treatment Centre, parasitological status, Initial cure, non-response to treatment, slow response to treatment, run away from treatment center, length of hospitalization and death.

### Sampling strategy

The household sample was estimated by taking into account the prevalence of VL in a similar setting [6] relative to the population of Moroto district. The household (HH) average size was considered as 5 members by household (Uganda Demographic Health Survey (UDHS) 2016) [9].

### Sample size

The participants were sampled in a multistage cluster sampling. The final cluster were the manyatta and by using a formula of cluster surveys, a total of 12 clusters (manyatta) were used for this study as used elsewhere [6,10].

$C = \frac{p(1-p)D}{s^2 b}$ Where, C: number of clusters selected

p: estimated prevalence of VL in Moroto district (p = 17.2% as per Odoch's study)

D: Design effect of 1.3 was used in this study because it has been used in a similar study in Amudat district, which neighbors Moroto district [6]. It has also a power of 0.7 [11].

b: Number of respondent per cluster (manyatta) and 24 was used in this study.

s: Standard error,

$s = \left(\frac{CI}{C_\alpha}\right)$ Where;

CI width: width of the desired confidence interval of 0.05

Cα: the value from the standard normal curve corresponding to the accepted alpha value of 1.96.

$C = \frac{0.172(1-0.172)1.3}{\left(\frac{0.05}{1.96}\right)^2 24}$ Sample size = C×b

This gives 288 participants.

## Sampling procedure

Moroto District has 3 counties, 8 sub counties and 39 parishes. According to the settings in Karamoja, there is often limited variation between counties and sub counties within the same district. Therefore, we considered parishes as our first stage of sampling, followed by the village and the households / individuals. We, therefore, applied multistage sampling. At the first stage, we randomly sampled 12 out of 39 parishes. Parishes were estimated to have between 3 and 7 manyattas. From each parish, we randomly sampled 1 Manyatta and from each Manyatta, 24 participants were sampled using systematic Sampling. The simple random sampling process for selection of parishes and Manyatta was done using ENA open software using. ENA has the facility to do random number selection, like other software such as Microsoft Excel. The Uganda Bureau of Statistics estimates that, on average, Manyatta population was 550 people as per 2020 data [12]. In our case, systematic sampling was done by lining up all participants available in each randomly selected manyatta at the time of the study, counting them and dividing their total number by 24 to get the sampling interval. Overall, we interviewed 288 respondents.

## Inclusion criteria

The specific criteria for inclusion of participants in this study was all individuals in the randomly selected households who were aged three months and above given that they were residents of the household at the time of conducting the study.

## Exclusion criteria

The exclusion criteria were any individual in the randomly selected household who met the inclusion criteria but were reported to be mentally ill.

## VL infection outcome

VL infection outcomes included:—Referred for treatment, Initial cure, non-response, slow response, defaulting, deaths and length of hospitalization. The study team visited Amudat Hospital for extraction of patients records (for patients previously referred from the community) using unique identifiers to determine the outcomes of the VL infection.

## Data collection techniques

Data collection ran for two months. After meeting the selected participants to take part in the study, the research team introduced themselves, explained the purpose of the study and sought for written consent from all participants. For child participants, written formal consent was obtained from their parents/guardians. After receiving their consent, the research team administered the questionnaires to the participants. The researcher checked for completeness of questionnaires before the participants leaving the study site. Completing a questionnaire took about 30 to 40 minutes. The tool was developed in reference to the earlier reviewed literature [8] and was improved using consultations from experts to help capture all the required data. Data was collected using structured researcher-administered questionnaires. This was specially designed in Open Data Kit (ODK) kobo collect to have check questions to ensure precision and accuracy. The signs and symptoms of VL for patients who reached the treatment Centre were assessed by ticking against a list of common VL symptoms and signs, which are in a form available at the treatment Centre.

## Data analysis

The overall analysis was conducted using STATA version-14 at 95% confidence level. Results have been presented using tables with their associated frequencies, percentages, odds ratios, confidence intervals and p-values where appropriate. Numerical data has been summarized into descriptive statistics of mean, median, range, Interquartile range (IQR) and standard deviations while categorical data was summarized into frequencies and percentages.

The predictors like the socio-demographic predictors of VL infection were collected and cleaned to form part of the analytic data set. Data was analyzed based on following procedures detailed below:- Data was analyzed using STATA version 14. The results were described using measures of central tendency, dispersion, and associations. For continuous variables, data was analyzed and presented as mean (SD), median (IQR) while for all categorical variables, data was analyzed and presented as percentages in tables and pie charts. The outcome variable of this study was prevalence of VL; which was computed as the proportion of participants who tested positive using the rK39 Rapid Diagnostic Test, divided by the total number of respondents tested using the rK39 Rapid Diagnostic Test. For factors associated with VL infection, the potential factors included socio-demographic characteristics, history of treatment for VL and spraying of animals with insecticides. To determine factors associated with VL infection, bivariate and multivariable logistic regression was conducted. Bivariate analysis was performed using chi-square. The level of confidence was set at 95%. In cases, where at-least a cell had a count of 5 or less, Fisher's exact test was used to test for significant association between the outcome and a given independent variable. A conservative level of significance set at $p<0.25$ was used for selecting independent variables to put into the multivariable logistic regression model. Multi-collinearity was checked for by conducting a correlation matrix among independent variables. A parsimonious multivariable logistic regression model was constructed to identify factors associated with VL infection among participants in Moroto district. Probability values of $< 0.05$ were significant. Results have been presented as frequencies, Odds ratios or probability values.

During both binary logistic and multivariate logistic regression analysis, clustering was factored in at county/constituency level of the participants, to cater for heterogeneity amongst participants in the three different counties. To factor in clustering at the county level during binary logistic and multivariate logistic regression analysis, the following steps were used; data structure was understood, a suitable model was selected, and we incorporated clustering. A mixed-effects **model was used** to cater for random effects for counties, which account for variation at the county level.

## Result

### Distribution of Respondents /Participants socio-demographic information

It is well known that Catholic is the predominant religion in Karamoja region, therefore, not surprising that most of the respondents were Catholics (93.40%). Of the respondents, 83.33% were married. In Moroto district, the predominant ethnic group are Mathenikos, who contributed to 65.63% of the respondents. Most respondents had never attended school (87.15%) and lived on an estimated monthly income of less than 5,000/ = (76.74%) as shown in Table 1.

### Prevalence of visceral leishmania infection in Moroto district

The study was carried out amongst 288 participants from Moroto district distributed as follows; 116 from Matheniko county, 89 from Tepeth county and 83 from Moroto Municipality

**Table 1. Socio-demographic Characteristics.**

| Variable | Frequency (n = 288) | Percentage |
|---|---|---|
| Age of Participants/years | | |
| 1/4–5 | 59 | 20.49 |
| 6–11 | 37 | 12.85 |
| 12–17 | 24 | 8.33 |
| 18–23 | 47 | 16.32 |
| 24–29 | 19 | 6.60 |
| 30–35 | 20 | 6.94 |
| Above 35 | 82 | 28.47 |
| Age of respondents/years | | |
| 13–19 | 6 | 2.00 |
| 20–35 | 161 | 55.90 |
| >35 | 121 | 42.01 |
| Sex of Participant | | |
| Female | 173 | 60.07 |
| Male | 115 | 39.93 |
| Religion of Respondents | | |
| Catholics | 269 | 93.40 |
| Anglicans | 6 | 2.08 |
| Moslems | 3 | 1.04 |
| Others | 10 | 3.47 |
| Marital status of respondents | | |
| Single | 34 | 11.81 |
| Married | 240 | 83.33 |
| Divorced | 3 | 1.04 |
| Widowed | 11 | 3.82 |
| Tribe of respondents | | |
| Matheniko | 189 | 65.63 |
| Tepeth | 94 | 32.64 |
| Bokora | 5 | 1.73 |
| Education level of respondents | | |
| None | 251 | 87.15 |
| Primary | 33 | 11.46 |
| Post-Primary | 4 | 1.39 |
| Nutritional Status of the Participant | | |
| Normal | 223 | 77.43 |
| Moderately Malnourished | 44 | 15.23 |
| Severely Malnourished | 21 | 7.29 |

based on the distributed of people in Moroto district as per the 2014 Uganda Population and housing Census report. Out of the 288 study participants, 5.21% (15/288) (95%CI: 3.15% - 8.48%) were found to have VL infection. The proportion of VL infection in Matheniko county, Tepeth county and Moroto Municipality was found to be 6.90% (8/116) (95%CI: 2.36% - 13.31%), 4.49% (4/89) (95%CI: 2.21% - 11.57%) and 3.61% (3/83) (95%CI: 2.06% - 10.85%) respectively. The distribution of VL infection across age group is as follows; Of the 15 individuals found to have VL, 6 (40.0%) were aged 3Months -17 years, 3 (20.0%) were aged 18–29 years, and 6 (40%) were aged 30 years and above.

### Factors associated with VL infection

During both binary logistic and multivariate logistic regression analysis, clustering was factored in at county/constituency level of the participants, to cater for heterogeneity amongst participants in the three different counties. The suitable model to explain factors associated with VL infection was based on either; (i) factors that were significantly associated with VL infection at bivariate level given that they showed no association with other independent variables or (ii) factors that showed no significant association with VL infection but were significantly associated with VL infection from other related studies and had P-value < 0.25 from the current study. As a result, nutritional status of the participants, presence of termite mounds around homestead, spraying of domestic animals in Kraals with insecticides, and sleeping under a mosquito net were included in addition to Malaria status of the participants and respondent's knowledge about; habitat for Sand flies, and transmission of VL infection. Nutritional status was assessed using Mid Arm Circumference measurements (MUAC). Knowledge about VL transmission was assessed by asking Participants about their knowledge on the way VL is transmitted. If they chose the option of a bite by a sandfly then we assumed, they were knowledgeable about transmission. Those who ticked otherwise were not knowledgeable.

Three feasible models were constructed and the Model with the lowest Akaike's Information Criteria (AIC) and Bayesian Information Criteria (BIC) was chosen as the best model fit to explain the factors associated with VL infection. The best model consists of factors shown in **Table 2**.

VL infection was significantly associated (since their p-values were less than 0.05 at Multivariate analysis level) with.

i. Respondent's knowledge about habitat for sand flies,

**Table 2. Factors associated with VL. Infection.**

| Variable | Visceral Leishmaniasis Infection | | COR (95% CI) | AOR, (95% CI) |
|---|---|---|---|---|
| | No, n (%) | Yes, n (%) | | |
| **Spraying domestic animals with Insecticides in Kraals** | | | | |
| **Yes** | 200(96.15) | 8(3.85) | 1.00 | 1.00 |
| **No** | 73(91.25) | 7(8.75) | 2.40[0.512–11.230] | 1.06[0.30–3.76] |
| **Respondent's Knowledge on transmission of VL.** | | | | |
| **Knows** | 155(97.48) | 4(2.52) | 1.00 | 1.00 |
| **Do not know** | 118(91.47) | 11(8.53) | 3.61[0.71–18.47] | 4.28[0.91–20.08] |
| **Respondent's Knowledge on habitat for Sand flies** | | | | |
| **Knows** | 155(98.10) | 3(1.90) | 1.00 | 1.00 |
| **Does not know** | 118(90.77) | 12(9.23) | 5.25[1.63–16.97]* | 5.33[1.69–16.82]** |
| **Presence of termite mounds around homesteads** | | | | |
| **Yes** | 232(95.47) | 11(4.53) | 1.00 | 1.00 |
| **No** | 41(91.11) | 4(8.89) | 2.06[0.76–5.59] | 1.27[0.25–6.55] |
| **Nutritional Status of the Participant** | | | | |
| **Normal** | 213(95.52) | 10(4.48) | 1.00 | 1.00 |
| **Moderately Malnourished** | 42(95.45) | 2(4.55) | 1.01[0.11–9.23] | 1.17[0.17–7.86] |
| **Severely Malnourished** | 18(85.71) | 3(14.29) | 3.55[2.23–5.66] *** | 3.12[0.83–11.74] |
| **Participant's Malaria status** | | | | |
| **Negative** | 136(97.84) | 3(2.16) | 1.00 | 1.00 |
| **Positive** | 137(91.95) | 12(8.05) | 3.97[0.81–19.47] | 3.86[0.98–15.17] |

Note: * p <0.05, **p<0.01, ***p<0.001, COR; Crude Odds Ratio, AOR; Adjusted Odds Ratio

The odds of having VL infection were 5.33 times higher among respondents who had incorrect knowledge of habitat for Sand flies as compared to those who had correct knowledge (AOR 5.33 (95%CI: (1.69–16.82).

## Outcomes of VL infection in Moroto district

Out of the 15 VL infected people who were followed up at Amudat Kala azar treatment center, 73.33% (11/15) (95%CI: 42.93% - 90.95%) were found to have reached the VL treatment center. The 11 patients who reached the treatment Centre, presented with following symptoms:— fever (81.82%), headache (72.72%), abdominal swelling (36.36%), abdominal pain (72.72%), diarrhea (36.36%), night sweats (63.63%), fatigue (63.63%), cough (63.63%), breathlessness (45.45%), and nose bleeding (9.09%).

All those who reached were hospitalized and never died or had a disability as a result of confirmed VL diagnosis. Out of the 11 people who reached the hospital, 72.72% (8/11) were parasitologically diagnosed of VL disease using bone marrow smear. Out of the 8 VL patients, 37.5% (3/8) were treated using SSG/PM which is a first line treatment, 62.5% (5/8) were recommended to get AmBisome, which is a second line treatment. However, only 1out of the 5 received AmBisome medication as the drug had run out of stock.

## Discussion of results

In this study, we found the prevalence of VL infection in Moroto district to be 5.21%. However, a review of admission records at Amudat hospital in 2017 showed that 27.1% were from Moroto district, the second largest proportion after Amudat (70.9%) indicating that Moroto District is one of the leading contributors of VL admissions at this hospital. For the first time we have documented community prevalence of 5.21% in Moroto district, representing a big burden of VL infection, because these are ordinarily undetected. These findings highlight a problem of this neglected disease in Northeastern Uganda. Analysis of the above VL admission proportions, indicate that the ratio of VL patients from Amudat district is 2.6 times higher compared to Moroto district. A study done by Odoch in 2010 revealed the VL infection prevalence of 17.2% in Amudat district [6], hence one would expect a VL infection prevalence of about 6.6% in Moroto district. However, our data showed a VL infection prevalence of 5.2% in Moroto district. These findings revealed that the VL infection prevalence in Moroto district varied from 6.90% in Matheniko County, to 4.49% in Tepeth County and to 3.61% in Moroto Municipality. The difference between our findings and the expected VL prevalence could be attributed to the difference in the sensitivity of the two tests used to detect VL infection. Whereas our study used rk39 to test for VL infection, Odoch's study used direct agglutination *test (DAT)*.It should be noted that when testing for VL infection, direct agglutination *test (DAT)* has a higher sensitivity of 92.0%, compared to rk39 whose sensitivity is 76.8% [4, 13]. Therefore, our study could have had a relatively a higher degree of false negatives compared to Odoch's study. Furthermore, according to Amudat hospital records, VL control measures like active case finding, diagnosis and treatment have been intensified in the last 9 years courtesy of Drugs for Neglected Diseases *initiative* (DND*i*) since Odoch's study was conducted in Karamoja region, inclusive of Moroto district.

In this study, children, constituted 40.0% of the VL infection burden. Those aged 35 years and above contributed 33.33% of the VL infection burden. It is important to note that Odoch's study was carried out in March, which is usually a rainy season in Uganda, when pastoralists are around their homesteads, hence the study was able to capture most of population at risk of VL infection (herdsmen; who are mostly aged between 6 and 17 years) [4]. However, the current study was carried out in December/January, which is a dry season in Uganda, when most

of these at-risk population of acquiring VL infection, were away from their homesteads in search of pasture and water for their animals. Therefore, they were not captured as revealed by our results (majority of our participants were below 5 years of age and those who were more than 35 years of age).

The sex ratio of VL infection was higher among females in a female to male sex ratio of 3:2, which contradicts earlier findings of a sex ration in female to males of 1:3 [4].

This change may indicate evolving epidemiological patterns influenced by various factors, including environmental, social, and behavioral dynamics.

For instance, the increased prevalence among females could be attributed to heightened exposure due to differing roles in household and agricultural activities, particularly in endemic regions. Additionally, it raises important questions about the biological and immunological factors that may predispose women to a higher infection rate in this context.

Further research is needed to explore these dynamics in depth and assess how such changes might affect public health strategies aimed at managing and preventing VL. Understanding the reasons behind this shift could lead to more effective interventions tailored to the specific needs of different populations. Overall, this emerging trend underscores the importance of continuously monitoring gender-specific health outcomes in infectious diseases.Our findings show that incorrect knowledge about the habitat for Sandflies increased the likelihood of VL infection. Perhaps, this could be because knowledge about the habitat for Sandflies means knowledge of VL transmission which forms a major pillar for people to guard themselves against acquisition of VL infection. Findings from this study are supported by findings from other studies that have assessed the association between knowledge of VL and VL infection in similar contexts. For example, a case-control study carried out in December 2006 in Pokot territory of Kenya and Uganda found that knowing the signs and symptoms of VL were associated with a protective effect [14]. People with incorrect knowledge about VL infection remain untreated which lead to a reservoir of the infection in the community.

Given that malaria is known to impair the immune response, one might expect that individuals with malaria could be more susceptible to VL due to the compromised immune system [15]. The lack of statistical significance in this association does not diminish its public health relevance. It may reflect underlying complexities in the interactions between these diseases. For example, individual variability in immune responses, age, sex, or co-existing health conditions could influence susceptibility and disease progression [16]. Additionally, the timing and severity of malaria infections could affect how the immune system responds to leishmaniasis, potentially obscuring clearer associations.

Environmental factors prevalent in Moroto, such as vector habitats and socio-economic conditions, may also contribute to the observed relationship. Areas with high malaria transmission often share ecological features with those conducive to VL transmission, which could result in overlapping risk factors [17]. Furthermore, behavioral aspects, such as outdoor activities that increase exposure to both vectors, may enhance the likelihood of co-infection. The study findings showed that nutritional status was not significantly associated with VL. This could be attributed to fact that malnutrition decreases the likelihood of diagnosis of asymptomatic VL infection. This indicates that malnutrition can impact the clinical presentation and detection of visceral leishmaniasis (VL) in individuals who do not show overt symptoms. Malnourished individuals often experience impaired immune function, which can lead to atypical or milder presentations of infections, making asymptomatic cases less likely to be identified. Furthermore, malnutrition may affect health-seeking behavior, as individuals with limited resources might not pursue medical attention unless symptoms are severe. Research has shown that malnutrition compromises the immune system, reducing the body's ability to mount effective responses to infections, which may alter the detection rates of asymptomatic

cases of VL [18]. This underscores the need for heightened awareness and targeted diagnostic approaches in malnourished populations to improve identification and management of asymptomatic VL. In Karamoja, the malnutrition levels have been on the rise in the immediate past 8 years as the Global Acute Malnutrition levels for Karamoja region have been worsening from 12.8% in 2014 to 13.8% in 2017 [19].

Though spraying animals with insecticides in Kraals was significantly associated VL in studies done outside Uganda [14], the current study findings show no significant association between VL infection and spraying of animals with insecticides. This is in agreement with a study done in Brazil in 2014 by Werneck to to determine the effectiveness of Insecticide Spraying and Culling of Dogs on the Incidence of Leishmania infantum Infection in Humans [20]. The discrepancy in the findings could be attributed to the fact that in the current study 72.2% of the respondents had their animals sprayed with insecticides, thus the data was skewed as far as spraying of animals with insecticides was concerned. This paused an analytical challenge in the comparison of the likelihood of acquiring VL infection among those who sprayed and those who never sprayed their animals with insecticides.

No significant association was found between having termite mounds around the homestead and VL, because majority of the household had termite mounds (84.38%), so the likelihood of having VL among those with and those without termite mounds could have been obscured. The current study findings are similar to findings from other studies reviewed [6,14].

The outcomes of VL infection indicates that majority of the infected VL patients (73.3%) sought health care. This could be attributed to the free transport to the VL treatment center at Amudat VL treatment Centre, free feeding and treatment supported by DNDi. Out of the 11 people who reached the hospital, 72.72% (8/11) were parasitologically diagnosed of VL disease using bone marrow smear. Out of the 8 VL patients, 37.5% (3/8) were treated using SSG/PM which is a first line treatment, 62.5% (5/8) were recommended to get AmBisome, which is a second line treatment [4]. However, only 20.0% (1/5) received AmBisome medication as the drug had run out of stock. All those who reached the facility and received treatment cured and were discharged. Those who had not yet received treatment were still on the ward waiting for second line treatment.

Though some studies register poor treatment outcomes arising from VL treatment [21], our findings showed only good treatment outcome as all those who received VL patients treatment were cured and discharged.

## Study limitation

The study faced two main limitations: incomplete follow-up and treatment availability issues. A significant portion of VL-infected individuals (26.67%) did not reach the treatment center, which may introduce bias in assessing treatment outcomes, as non-responders' outcomes could differ from those who sought treatment. Additionally, stockouts of AmBisome (a second-line treatment) at the treatment center limited its availability to some patients, highlighting gaps in healthcare infrastructure that could affect the generalizability of the treatment results to other settings.

## Conclusion

Visceral Leishmaniasis or Kala-azar neglected tropical disease is still prevalent in Moroto district, with a prevalence of 5.21%. County distribution of VL infection ranged from 6.90% in Matheniko County, 4.49% in Tepeth County to 3.61% in Moroto Municipality. From the results, the high-risk factor for VL infection in Moroto district was noted to be incorrect

knowledge of habitat for Sand flies. The treatment outcome for VL diagnosis included hospitalization of not more than 17 days, and all the treated patients cured. The signs and symptoms present by VL infected individuals included fever, headache, abdominal pain and swelling, coughing, night sweats, diarrhea, fatigue, breathlessness, and nose bleeding. We recommend that VL infection be controlled and plan for its elimination.

## Supporting information

**S1 Data. Baseline Infection Prevalence, Risk Factors and Treatment Outcomes of Visceral Leishmaniasis in Northeastern Uganda.**
(DTA)

**S1 File. Variable Names and Descriptions.**
(DOCX)

## Author Contributions

**Conceptualization:** Patrick Sagaki, Peter Olupot-Olupot.

**Data curation:** Patrick Sagaki, Posiano Mulalu, Peter Olupot-Olupot.

**Formal analysis:** Patrick Sagaki, Posiano Mulalu, Peter Olupot-Olupot.

**Funding acquisition:** Patrick Sagaki.

**Investigation:** Patrick Sagaki, Posiano Mulalu, Peter Olupot-Olupot.

**Methodology:** Patrick Sagaki, Posiano Mulalu, Peter Olupot-Olupot.

**Project administration:** Patrick Sagaki.

**Resources:** Patrick Sagaki.

**Software:** Patrick Sagaki, Posiano Mulalu.

**Supervision:** Patrick Sagaki, Posiano Mulalu, Benon Wanume, Peter Olupot-Olupot.

**Validation:** Patrick Sagaki, Posiano Mulalu, Peter Olupot-Olupot.

**Visualization:** Patrick Sagaki, Posiano Mulalu, Peter Olupot-Olupot.

**Writing – original draft:** Patrick Sagaki, Posiano Mulalu.

**Writing – review & editing:** Patrick Sagaki, Jeremiah Mutwalante Twa-Twa, Posiano Mulalu, Benon Wanume, Peter Olupot-Olupot.

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
