## [Decision Letter · Decision Letter 0]

16 Sep 2024

Dear Dr Sagaki,

Thank you very much for submitting your manuscript "Pre-elimination Prevalence, Risk Factors and Treatment Outcomes of Visceral Leishmaniasis in Northeastern Uganda: A cross-sectional study" for consideration at PLOS Neglected Tropical Diseases. As with all papers reviewed by the journal, your manuscript was reviewed by members of the editorial board and by several independent reviewers. In light of the reviews (below this email), we would like to invite the resubmission of a significantly-revised version that takes into account the reviewers' comments. Please ensure that all you provide a comprehensive and complete answer to each of the the reviewers' comments.

We cannot make any decision about publication until we have seen the revised manuscript and your response to the reviewers' comments. Your revised manuscript is also likely to be sent to reviewers for further evaluation.

Sincerely,

Shaden Kamhawi

Editor-in-Chief

Reviewer's Responses to Questions

**Key Review Criteria Required for Acceptance?**

**Methods**

-Are the objectives of the study clearly articulated with a clear testable hypothesis stated?

-Is the study design appropriate to address the stated objectives?

-Is the population clearly described and appropriate for the hypothesis being tested?

-Is the sample size sufficient to ensure adequate power to address the hypothesis being tested?

-Were correct statistical analysis used to support conclusions?

-Are there concerns about ethical or regulatory requirements being met?

Reviewer #1: There are several issues regarding methods that should be addressed, please see "Summary and General Comments"

Reviewer #2: 1- The objectives of the study did not clearly articulate with a clear testable hypothesis.

2- The study designed appropriate.

3- Population clearly described.

4- The sample size sufficient to ensure adequate power.

5- Correct statistical analysis used to support conclusions.

6- There were not concerns about ethical or regulatory requirements being met.

Reviewer #3: -Are the objectives of the study clearly articulated with a clear testable hypothesis stated? Yes

-Is the study design appropriate to address the stated objectives? Yes

-Is the population clearly described and appropriate for the hypothesis being tested? No

-Is the sample size sufficient to ensure adequate power to address the hypothesis being tested? Yes

-Were correct statistical analysis used to support conclusions? Yes

-Are there concerns about ethical or regulatory requirements being met? No

**Results**

-Does the analysis presented match the analysis plan?

-Are the results clearly and completely presented?

-Are the figures (Tables, Images) of sufficient quality for clarity?

Reviewer #1: There are several issues regarding results that should be addressed, please see "Summary and General Comments"

Reviewer #2: 1- The analysis presented matched the analysis plan.

2- The results were presented clearly and completely.

3- The figures (Tables, Images) were sufficient for clarity.

Reviewer #3: -Does the analysis presented match the analysis plan? No

-Are the results clearly and completely presented? No

-Are the figures (Tables, Images) of sufficient quality for clarity? Yes

**Conclusions**

-Are the conclusions supported by the data presented?

-Are the limitations of analysis clearly described?

-Do the authors discuss how these data can be helpful to advance our understanding of the topic under study?

-Is public health relevance addressed?

Reviewer #1: Conclusions are not adequate, it just describes a summary of the results and their implications for increasing knowledge and informing public health actions.

Reviewer #2: 1- The conclusions were supported by the data presented.

2- The limitations of analysis were not described.

3- The authors did not discuss how data can be helpful to advance our understanding of the topic under study.

4- The public health relevance was partly addressed.

Reviewer #3: -Are the conclusions supported by the data presented? No

-Are the limitations of analysis clearly described? No

-Do the authors discuss how these data can be helpful to advance our understanding of the topic under study? yes

-Is public health relevance addressed? Yes

**Editorial and Data Presentation Modifications?**

Reviewer #1: Please see "Summary and General Comments"

Reviewer #2: Major Revision

Reviewer #3: (No Response)

**Summary and General Comments**

Reviewer #1: The cross-sectional study aims to describe the pre-elimination prevalence, associated factors, and treatment outcomes of VL in Moroto District, Uganda. Some major issues need to be considered for revision.

- Abstract: In the results section, authors declare that "The average hospitalization for VL was 17 days", implying that some patients have longer lengths of stay. However, in the conclusion, they say, "Patients with VL were hospitalized for not more than 17 days". Please clarify such inconsistency.

- Introduction, page 3, line 4: Remove the phrase "in areas where studies have been conducted."

- Introduction, page 3, lines 5/6: This set of countries no longer accounts for 95% of cases worldwide; the reference is outdated. Kenya and Somalia are among the most affected countries today, while Bangladesh and Nepal are out.

- Introduction, page 3, lines 10/11: certainly more than than ref; referencestoo old.

The authors use the term "pre-elimination prevalence" in the title, abstract, and throughout the text, suggesting that some effort is being made to eliminate VL from this region. However, there is no evidence in the text that a formal elimination program is ongoing. Therefore, it would be better to refer only to "prevalence" or "baseline prevalence."

The parameters used for sample size selection need to be given. The author presents a formula for the number of clusters, fixing the number of 24 respondents in each cluster. We need to know the estimated prevalence of VL in the Moroto district (p) to evaluate whether the final sample size (n=288) is adequate. 

- Regarding sample size, there is also a need for a better justification for using a design effect of 1.3. Authors say this value was "used in similar study design of VL in a community survey in Uganda (Odoch & Olobo 2013)." Consulting such a study, we can see that they used the value of D=1.3 based on a study in a cluster randomized trial in Brazil. There is no reason to believe that the value calculated in a different context would apply in the Moroto district. With the actual results of this study, the author may check if such an assumption is correct. 

- Data collection techniques: Please remove the first paragraph. This is unnecessary in this context. 

- Data analysis: Please make sure to indicate if sampling weights were used to correct estimates for complex sampling procedures. Explain how "clustering was factored in at county/constituency level" was done. Did you use multilevel or GEE approaches?

- Data analysis: please make clear what does "(plus the ones I suggested)" mean.

- Ethics: Please indicate the number of ethical approval documents and submit a copy. 

- Please indicate how nutritional status and knowledge of transmission was assessed. 

- Discussion, 3rd paragraph: The fact that you had more females in the sample cannot explain the higher sex ratio of leishmania infection among women because prevalences are calculated conditional on sex. 

- Discussion, malaria: explanation is not adequate. You found a strong association between positivity to malaria and the prevalence of infection, although not significant (probably due to the insufficient statistical power of the study). 

- Discussion, malnutrition: Please explain and provide a reference for the sentence "malnutrition decreases the likelihood of diagnosis of asymptomatic VL infection."

Reviewer #2: (No Response)

Reviewer #3: In this manuscript the the authors evaluated the Prevalence, Risk Factors and Treatment Outcomes of Visceral Leishmaniasis in Northeastern Uganda. TThe situation discussed is of utmost importance to better understand how VL behaves in different regions. However, the manuscript needs to be greatly improved to make it more robust and with more precise information about the topic addressed. Here are my considerations.

1) Title. I suggest you review the title, as I believe that the current format (Pre-elimination...) does not make it clear that the disease was subsequently eliminated or if any action was implemented to that end.

2) Abstract

The authors mention that patients with VL presented the following symptoms: fever, headache, abdominal pain and swelling, coughing, night sweats, diarrhea, fatigue, breathlessness, and nose bleeding. However, this information is not included in the results of the manuscript. 

3) Methodology: 

a) It is unclear how malnutrition was measured.

b) It is unclear how symptoms and signs of the disease were assessed.

4) Results. The results presented are a bit confusing. Some numbers do not agree. I suggest reviewing point by point as follows:

a) Some data are not in agreement. The authors report that there were 34 single people, however only in the age group of 1 to 4-5 years were 59 participants included and in the age group of 6 to 11 years 37 participants were included. I suggest reviewing the data or the way of writing these results.

b) Regarding the prevalence of infection, which age group was most affected? This is not clear. Furthermore, participants with positive results presented some symptoms of the disease. This is described in the abstract, but this information is not included in the results. Regarding the prevalence of infection, I ask whether patients were tested for HIV infection. This information is important, since it is known that serological tests have lower sensitivity in this population.

PLOS authors have the option to publish the peer review history of their article (what does this mean?). If published, this will include your full peer review and any attached files.

Reviewer #1: No

Reviewer #2: No

Reviewer #3: No
---

## [Editor Report · Decision Letter 1]

12 Nov 2024

PNTD-D-24-00638R1Pre-elimination Prevalence, Risk Factors and Treatment Outcomes of Visceral Leishmaniasis in Northeastern Uganda: A cross-sectional studyPLOS Neglected Tropical Diseases Dear Dr. Sagaki, Thank you for submitting your manuscript to PLOS Neglected Tropical Diseases. After careful consideration, we feel that it has merit but does not fully meet PLOS Neglected Tropical Diseases's publication criteria as it currently stands. Therefore, we invite you to submit a revised version of the manuscript that addresses the points raised during the review process. Please submit your revised manuscript within 30 days Dec 12 2024 11:59PM. If you will need more time than this to complete your revisions, please reply to this message or contact the journal office at plosntds@plos.org. Please include the following items when submitting your revised manuscript:*
A rebuttal letter that responds to each point raised by the editor and reviewer(s). You should upload this letter as a separate file labeled 'Response to Reviewers'. This file does not need to include responses to any formatting updates and technical items listed in the 'Journal Requirements' section below.*
A marked-up copy of your manuscript that highlights changes made to the original version. You should upload this as a separate file labeled 'Revised Manuscript with Track Changes'.*
An unmarked version of your revised paper without tracked changes. You should upload this as a separate file labeled 'Manuscript'. If you would like to make changes to your financial disclosure, competing interests statement, or data availability statement, please make these updates within the submission form at the time of resubmission. Guidelines for resubmitting your figure files are available below the reviewer comments at the end of this letter. We look forward to receiving your revised manuscript. Kind regards, Shaden KamhawiEditor-in-ChiefPLOS Neglected Tropical Diseases Shaden Kamhawi

co-Editor-in-Chief

Paul Brindley

co-Editor-in-Chief

 **Journal Requirements:** **Additional Editor Comments (if provided):** Please change the title to 'Baseline infection prevalence' instead of Pre-elimination unless there are concrete and official plans at the government level to eliminate VL from Northeastern Uganda

Add a paragraph on the limitation of the study in the discussion**Reviewers' comments:**   **Figure resubmission:** While revising your submission, please upload your figure files to the Preflight Analysis and Conversion Engine (PACE) digital diagnostic tool, https://pacev2.apexcovantage.com/. PACE helps ensure that figures meet PLOS requirements. To use PACE, you must first register as a user. Registration is free. Then, login and navigate to the UPLOAD tab, where you will find detailed instructions on how to use the tool. If you encounter any issues or have any questions when using PACE, please email PLOS at figures@plos.org. Please note that Supporting Information files do not need this step. If there are other versions of figure files still present in your submission file inventory at resubmission, please replace them with the PACE-processed versions. **Reproducibility:** To enhance the reproducibility of your results, we recommend that authors of applicable studies deposit laboratory protocols in protocols.io, where a protocol can be assigned its own identifier (DOI) such that it can be cited independently in the future. Additionally, PLOS ONE offers an option to publish peer-reviewed clinical study protocols. Read more information on sharing protocols at https://plos.org/protocols?utm_medium=editorial-email&utm_source=authorletters&utm_campaign=protocols

---

## [Editor Report · Decision Letter 2]

16 Dec 2024

Dear Dr Sagaki,

We are pleased to inform you that your manuscript 'Baseline Infection Prevalence, Risk Factors and Treatment Outcomes of Visceral Leishmaniasis in Northeastern Uganda: A cross-sectional study' has been provisionally accepted for publication in PLOS Neglected Tropical Diseases.

Best regards,

Shaden Kamhawi

Editor-in-Chief

Shaden Kamhawi

co-Editor-in-Chief

Paul Brindley

co-Editor-in-Chief

---

## [Editor Report · Acceptance letter]

4 Jan 2025

Dear Dr Sagaki,

We are delighted to inform you that your manuscript, "Baseline Infection Prevalence, Risk Factors and Treatment Outcomes of Visceral Leishmaniasis in Northeastern Uganda: A cross-sectional study," has been formally accepted for publication in PLOS Neglected Tropical Diseases.

Best regards,

Shaden Kamhawi

co-Editor-in-Chief

Paul Brindley

co-Editor-in-Chief
